# Episodic Angioedema with Hypereosinophilia (Gleich’s Syndrome): A Case Report and Extensive Review of the Literature

**DOI:** 10.3390/jcm10071442

**Published:** 2021-04-01

**Authors:** Ilaria Mormile, Angelica Petraroli, Stefania Loffredo, Francesca Wanda Rossi, Mauro Mormile, Andrea Del Mastro, Giuseppe Spadaro, Amato de Paulis, Maria Bova

**Affiliations:** 1Department of Translational Medical Sciences and Center for Basic and Clinical Immunology Research (CISI), WAO Center of Excellence, University of Naples Federico II, Via S. Pansini 5, 80131 Naples, Italy; ilariamormile@virgilio.it (I.M.); stefanialoffredo@hotmail.com (S.L.); frawrossi@yahoo.it (F.W.R.); spadaro@unina.it (G.S.); depaulis@unina.it (A.d.P.); bovamaria@virgilio.it (M.B.); 2Institute of Experimental Endocrinology and Oncology “G. Salvatore” (IEOS), National Research Council (CNR), Via S. Pansini 5, 80131 Naples, Italy; 3Department of Clinical Medicine and Surgery, University of Naples Federico II, Via S. Pansini 5, 80131 Naples, Italy; mormile@unina.it; 4Emergency Division, A.O.R.N. “Antonio Cardarelli”, Via Antonio Cardarelli, 9, 80131 Naples, Italy; andrea.delmastro@aocardarelli.it

**Keywords:** angioedema, acquired angioedema, episodic angioedema with eosinophilia, Gleich’s syndrome, hypereosinophilia, interleukin-5, urticaria

## Abstract

Episodic angioedema with eosinophilia (EAE) (Gleich’s syndrome) is a rare disease characterized by hypereosinophilia (up to 95 × 10^9^ cells/L), recurrent episodes of angioedema, urticaria, weight gain, and fever, that occur at periodical intervals (usually every 3–4 weeks). The exact etiology of EAE is still unclear, but both eosinophils and abnormalities of cytokines homeostasis seem to play a pivotal role in the pathogenesis of the disease. In particular, the cyclic elevation of serum interleukin-5 before the increase in eosinophil count has been reported. Herein, we performed a broad literature review and report the case of a thirty-two-year-old woman with a two-year history of cyclic angioedema attacks, urticaria, periodic weight gain, and severe hypereosinophilia, diagnosed with EAE and treated with oral corticosteroids. Describing the most relevant clinical features of EAE reported so far in the literature, we aim to provide physicians with some useful tools to help them deal with this disease. In addition, we aim to raise awareness about this rare condition in which approved diagnostic classification criteria are currently missing.

## 1. Introduction

Episodic angioedema with eosinophilia (EAE), also known as Gleich’s syndrome, is a rare disease characterized by hypereosinophilia (up to 95 × 10^9^ cells/L) and recurrent episodes of angioedema (AE), urticaria, weight gain, and fever, that occur at periodical intervals (usually every 3–4 weeks) [1,2,3,4]. Until now, fewer than 100 patients have been reported, mainly adults. A recent systematic review of the literature of all pediatric cases of EAE reported 17 cases of EAE occurring in children [5].

EAE has been classified in the broad category of hypereosinophilic syndromes (HES) [6]. Despite that, EAE patients usually show homogeneous clinical presentations and well-characterized laboratory findings; these considerations suggest that EAE should be considered a distinct eosinophilic disease that is likely caused by a common pathogenetic mechanism in affected subjects [3].

The exact etiology of EAE is still unclear. However, both eosinophils and abnormalities of cytokine homeostasis seem to play a pivotal role in the pathogenesis of the disease, especially the cyclic elevation of serum interleukin (IL)-5 before the increase in eosinophil count has been reported [7,8]. Furthermore, isolated case reports found other cytokine cyclic elevations (granulocyte macrophage colony-stimulating factor (GM-CSF), IL-3, IL-6, IL-1, and soluble IL-2 receptor), even though the sources of these cytokines have not yet been identified [9,10].

An increased number of activated T cells and eosinophilic degranulation has been described in the dermis during acute attacks, suggesting that the activation of blood and tissue eosinophils due to the T cell stimulation is likely the main mechanism involved in EAE pathogenesis [2,7,8]. In a recent study, cyclic variation of other peripheral cells (such as neutrophils, lymphocytes, and mast cells) was observed in four patients, suggesting that these cells could play a role in the pathogenesis of the disease [3].

Herein, we perform a broad literature review and report the case of a thirty-two-year-old woman with a two-year history of cyclic AE attacks and severe hypereosinophilia, which emphasizes how differential diagnosis may be challenging when dealing with a rare and little-known disease. Describing the most relevant clinical features of EAE reported so far in the literature, we aim to provide physicians with some useful tools to help them deal with this disease. In addition, we aim to raise awareness about this rare condition in which approved diagnostic classification criteria are currently missing.

## 2. Patients and Methods

### 2.1. Case Report

A thirty-two-year-old female accessed our Immunology Clinic reporting a history of cyclical attacks of AE, urticaria, and periodic weight gain (up to 10 kg) during the previous two years that caused several visits to the Emergency Room. The attacks mainly involved the face, arms, and left leg, and were sometimes associated with mild itching. They occurred approximately weekly and showed a good response to high doses of oral glucocorticoids. The edema lasted approximately 3–4 days if not treated. The patient’s past and family medical history were negative for AE episodes, allergic disorders, autoimmune diseases, and other common eosinophilic diseases. She reported that, in the previous six months, laboratory investigations had shown a recurrent increase of total leukocyte count (up to 23,400/mm^3^ during the acute phase), eosinophil levels (9617/mm^3^; 41%), and neutrophil count (9009/mm^3^; 38.50%). Hemoglobin and platelet counts had always been normal. The patient had already undergone several investigations, including blood chemistry, serum protein electrophoresis, immunological tests for connective tissue diseases, erythrocyte sedimentation rate (ESR), urine analysis, parasitology, and chest X-ray, which were all within normal limits. An abdominal ultrasound showed mild hepatomegaly and two concomitant small hepatic cysts (1 cm) and splenomegaly (splenic length 140 mm).

When the patient was referred to our care, unilateral periorbital edema of the face was evident on physical examination, her body temperature and other vital signs were within normal limits, and her body mass index (BMI) was 31.9 kg/m^2^. We repeated laboratory investigations that revealed a total leukocyte count of 17,200/mm^3^ with 45.5% of eosinophils (7830/mm^3^). Hemoglobin, neutrophil, and platelet counts were normal.

Because the patient had a history of rhinitis and chronic sinusitis, we assessed total IgE and specific IgE for most common grass and tree pollens, dust mites, animal dander, and molds, which were either in the normal range or were negative. A maxillofacial computed tomography (CT) scan showed unilateral maxillary sinusitis (Figure 1).

Other tests such as C-reactive protein (CRP) concentrations, rheumatoid factor, anti-nuclear antibodies (ANA), ANCA-MPO, PR3-ANCA, DSDNA-Ab, IgG, IgA, IgM, anti-transglutaminase IgA, C3, C4, C1-INH, C1Q, and tryptase levels were within normal limits. Tumor marker levels (AFP, CEA, CA 19-9, CA 15-3, CA 125, β2 microglobulin) were not increased. Multiple stool examinations and *“scotch tape”* tests were performed to detect parasites, resulting negative. IgE for *Echinococcus* was not found. Blood cell immunophenotyping identified: CD3, 83%; CD4, 54%; CD8, 14%; CD19, 7%; CD56, 7%; CD3^+^CD56^+^, 2%; CD25^+^CD3^+^, 1%; TCRαβ, 69%; CD3^+^CD4^-^CD8^-^TCRαβ^+^, 0.9%; and CD4%/CD8%, 3.86%.

No morphological abnormalities were detected on peripheral blood smear examination. Because the patient also complained of pyrosis and epigastric pain, she underwent an upper gastrointestinal endoscopy (Figure 2) plus random biopsies, which revealed mild esophagitis with no eosinophilic infiltrate.

Spirometry and echocardiography were normal, whereas a CT scan of the chest detected subcentimetric anterior mediastinal lymph nodes and axillary lymphadenopathy (maximum diameter 1.3 cm). Multiple small reactive lymphadenopathies were also detected at ultrasound examination of the cervical and inguinal regions and at the Positron Emission Tomography/Computed Tomography (PET/CT) total body scan (Figure 3), which also revealed a mild and homogenously diffuse 2-deoxy-2-[fluorine-18] fluoro-D-glucose (18F-FDG) bone marrow uptake.

Bone marrow aspiration and biopsy were performed to rule out any associated malignancy, which showed only an increase in eosinophilic precursors and mild T cell (CD5^+^, CD20^-^) infiltrate, but no atypical cells.

Genetic tests for mutations, including *BCR/ABL* and *FIP1L1-PDGFRA*, performed on both peripheral blood and bone marrow cells, were negative.

According to clinical criteria, the patient was diagnosed with EAE and she was treated with high doses of oral glucocorticoids (Prednisone 75 mg/day for two weeks) with symptoms resolution and normalization of eosinophil counts. Subsequently, an oral prednisone tapering was scheduled up to the current maintenance dose of 5 mg/day. At the last follow-up (12 months from the EAE diagnosis), the patient has achieved clinical and biological remission.

The clinical characteristics of the patient are summarized in Table 1.

### 2.2. Research Method

This review includes all articles from December 2020 and earlier, written in English, and published in peer-reviewed and international journals that dealt with EAE. We searched for original articles, case reports, and letters to the editor reporting adult and pediatric cases of EAE in an online medical database (PubMed). The keywords used for searching PubMed were “Gleich’s syndrome” and “episodic angioedema with eosinophilia”. Written informed consent was obtained from the patient we described herein.

## 3. Episodic Angioedema with Hypereosinophilia Clinical and Laboratory Features

The current clinical experience with EAE is limited to case reports and case series. Haber et al. [11] performed a systematic review of the literature, analyzing 31 case reports and series of EAE patients to extrapolate clinical, epidemiological, and therapeutic data on this syndrome. EAE seems to be more common in females with no specific age group for disease onset (mean age of onset: 34.75 years; range: 2–45 years of age) [11]. Most EAE cases have been identified in the United States, Europe, and Asia [5,11].

EAE patients may show a broad clinical spectrum (Table 2). Abisror et al. [12] tried to assess the frequency of each of the symptoms in a recent retrospective multicenter study that included 30 patients with EAE. According to the authors, the main features reported in EAE patients are AE attacks (80%), which mainly affect the face, the neck, and the upper limbs; episodic abdominal pain (33%); diarrhea (30%); vomiting (10%); concomitant weight gain (up to 10 Kilograms) (23%); arthralgia (23%); lymphadenopathy (23%); splenomegaly (23%); and asymptomatic eosinophilic myocarditis (7%) [12]. Weight gain is due to fluid retention; it may be sudden and increase up to 10–20% of the baseline weight [13]. Oliguria has also been reported [13]. The clinical picture of pediatric cases mimics that of adult patients [5].

Laboratory findings of patients with EAE are summarized in Table 3. The most common laboratory alteration is the elevation of eosinophil count. In a recent study, other peripheral cell cyclic variation (such as neutrophils, lymphocytes, and mast cells) has been reported, suggesting that EAE is a multilineage cell cycling disorder [3].

Increased IL-5 levels have been observed several days before the eosinophilia peak and during the attacks, with subsequent normalization [13]. Other frequent changes in laboratory parameters include high total serum IgM (67%) and IgE levels (63%), abnormal T-cell immunotypes (40%), and evidence of clonal TCR γ gene rearrangements (17%) [12]. The aberrant T cell immunophenotypes identified in EAE patients are: CD3^-^CD4^-^ clonal T cells (which are also common in the lymphocytic variant of HES) [9,24,25], CD3^-^CD4^+^, CD3^+^CD4^+^CD7^-^, CD3^-^CD3ic^+^CD4^-^CD8^-^CD7^+^CD5^+^, CD3^+^CD4^-^CD8^-^CD2^+^CD5^+^CD7^+^TCRγδ^+^ phenotype [12], and CD3^+^CD4^-^CD8^-^ lymphocyte population [15]. However, many patients do not show such T cell clones, although they might develop them during the course of the disease [1].

Because extensive EAE studies are currently lacking, it is difficult to assess long-term outcomes for these patients. Indeed, although EAE usually has a benign clinical course and is generally not associated with end-organ damage, the occurrence of neuritis and mitral valve insufficiency necessitating replacement have previously been reported [1].

Laboratory findings (such as eosinophilia, IgM or IgE levels, and the presence of aberrant T cell population) cannot be considered reliable predictors of flare incidence. However, abnormal T cell populations have been associated with a shorter time to relapse [12] and more frequent peripheral lymphadenopathy and urticaria [12]. It is not clear if patients affected by EAE with aberrant phenotypes can develop T-cell lymphoma; indeed, this association has previously been reported but not strictly correlated [12]. For all these reasons, further studies are needed to assess the long-term prognosis of EAE patients.

Histology is not mandatory to assess EAE diagnosis. However, skin biopsies may show some interesting features such as eosinophilic infiltrate and deposition of eosinophil granule proteins, together with an increased number of activated T cells and eosinophilic degranulation in the dermis during acute attacks [3,5].

Histopathological findings could also be useful to evaluate internal organ involvement due to eosinophilic infiltration. Gastrointestinal manifestations are some of the most frequent and debilitating aspects of HES [26]. The endoscopic features may include esophageal furrows, rings, plaques, mucosal erythema, and ulceration [27]. Gastroenterological symptoms in both HES with multiorgan system involvement and localized eosinophilic gastrointestinal disease may often be discordant with endoscopic or histopathologic appearance, as in the case of other inflammatory bowel diseases such as Crohn’s disease and ulcerative colitis [27]. In eosinophilic esophagitis, the endoscopic appearance of the esophagus may be normal in 10% to 25% of patients; for this reason, esophageal biopsies are recommended in patients in whom there is clinical suspicion of eosinophilic involvement of the esophagus, even if there are no endoscopic alterations of the esophageal mucosa [28,29]. Histological findings due to the eosinophil infiltration of the gastrointestinal tract may indeed be discontinuous, occurring in patches; therefore, random biopsies should be taken from the upper, middle, and lower thirds of the esophagus [21,23,30]. Because our patient complained of gastric symptoms, we performed upper gastrointestinal endoscopy with biopsies, observing mild esophagitis (Figure 2) with no eosinophilic infiltrate. Although gastrointestinal symptoms have been previously described in EAE, this is the first report of upper gastrointestinal endoscopy in an EAE patient; for these reasons, further studies are required to better elucidate the endoscopic and histologic gastrointestinal patterns of this disease. 

## 4. Differential Diagnosis

Due to the extreme rarity of the disease, there are no approved classification criteria for the diagnosis of EAE. The inclusion criteria for the assessment of diagnosis of EAE, which have been used by the broader retrospective case series that are available up to now, are the following: (I) episodic and recurrent AE, (II) concomitant blood hypereosinophilia (i.e., eosinophils higher than 1.5 G/L at least twice), and (III) the exclusion of an alternate diagnosis for both AE and eosinophilia [5,12,31].

Differential diagnosis is often a challenging process for physicians dealing with a rare disease. In Table 4 we summarized EAE main differential diagnosis.

IL-2-induced capillary leak syndrome presents high levels of both IL-5 and eosinophils [12,32]. In addition, because one of the most common presentation in EAE is the swelling affecting the face (80%) [12], this condition should be differentiated from hereditary (HAE) and acquired bradykinin-mediated AE (AAE) (for patients presenting with AE without urticaria), and from histamine-mediated AE (for patients experiencing concomitant pruritus and urticaria). An accurate familiar, personal, and drug anamnesis, together with blood testing for complement levels and genetic investigations, may help rule out these disorders. Moreover, no life-threatening attacks have been reported in EAE, and the swelling rarely involves visceral organs; these features could help distinguish EAE from HAE/AAE [33]. Despite this, a recent case report attributes abdominal pain and eosinophilic ascites to exacerbation of EAE [34].

EAE has been classified as a subgroup of HES [6]. Eosinophilia may be due to primary and secondary causes [35]. Primary (clonal) eosinophilia occurs in some hematological neoplasms in which eosinophils are part of the neoplastic clone [35]. Reactive hypereosinophilia occurs in many T-cell and B cell lymphoproliferative disorders; moreover, EAE has been described in lymphoid-HES in which mature peripheral T cells produce high amounts of IL-5, which stimulates eosinophil expansion [36,37]. Idiopathic hypereosinophilia is a diagnosis of exclusion in patients who have properly undergone a complete and detailed diagnostic work-up without detectable primary or secondary causes of eosinophilia being found [35].

Infectious diseases may cause both secondary hypereosinophilia and chronic spontaneous urticaria (CSU). Studies have demonstrated suppression of peripheral eosinophils count in patients during viral and bacterial infections [36,37], whereas parasitic and fungal (e.g., coccidomycosis) infections can elicit eosinophilia [38]. In industrialized countries, parasitic infections are a possible but uncommon cause of CSU as their prevalence in CSU patients varies greatly depending on the geographical area, climate, hygiene, and socio-economic conditions (adults CSU patients: 0–75.4%; pediatrics CSU patients: 0–37.8%) [39]. The concomitance of gastrointestinal symptoms, a history of previous parasitic infections, travels abroad, and unexplained eosinophilia should always be evaluated in CSU patients. However, each parasitic species and even different stages of parasite development may differently stimulate the host’s immune response and influence skin symptoms, as well as the level of increase of the eosinophil count (e.g., in *Blastocystis* spp. infection, the amoeboid form is the one most observed in patients with CSU) [40,41]. Helminths and protozoa are the parasites most commonly associated with CSU, but early helminth infections are often associated with pronounced eosinophilia, while protozoa, in general, do not cause eosinophilia [38,42].

Our patient was diagnosed with EAE based on the characteristic clinical features (i.e., recurrent episodes of urticaria, AE, and concomitant increased body weight, together with the remarkable elevation of eosinophils during the exacerbation). We tested the *FIP1L1/PDGFRA* and *BCR/ABL* fusion genes to exclude the possibility of a myeloid neoplasm or clonal eosinophilia. Moreover, the absence of concomitant allergic, infective, and connective tissue disorders, together with a significant clinical improvement (with a rapid decrease of the eosinophil count to normal levels) following corticosteroid therapy, was pivotal to assess the diagnosis. Our patient had normal immunoglobulin levels on admission, so we would like to stress that IgM levels are elevated in many, but not all EAE patients [4,5,11,16,17,18]. Laboratory findings are not pathognomonic in EAE, except for the typical elevation of the eosinophils count during attacks. However, a comprehensive diagnostic work-up through laboratory investigations, together with a detailed medical history, may be useful to address the differential diagnosis. Notably, differentiating EAE from HES is very important because prognosis and treatment strategies are very different. In addition, despite the elevation of eosinophils being a common feature of those diseases, underlying pathophysiological mechanisms (such as the peripheral blood eosinophil activation state) are possibly different. Indeed, Kawano et al. [16] reported that CD69, one of the surface antigens of activated eosinophils, was not expressed on the peripheral eosinophils of a woman diagnosed with EAE in contrast with HES. For all these reasons, although several distinguishing features can guide the diagnosis, EAE mainly remains a tentative diagnosis (Figure 4), and systematic classification criteria for this disease are needed.

## 5. Treatment Strategies and Future Perspective

EAE usually shows a benign course, presenting with no life-threatening attacks, differently from HAE and AAE [12,33], and lack of end-organ damage differently from HES [4]. The single attack may resolve spontaneously without therapy in 7–10 days and, generally, the disease shows a good response to systemic corticosteroids [2,3,11]. Indeed, EAE patients are usually managed with daily administration of oral glucocorticoids, with an increase of the dose (50–75 mg/day) during the acute phases of the disease [15]. However, although glucocorticoids can induce an initial remission of the disease in most cases, only a few patients can achieve sustained clinical and biological remission with fewer than 10 mg of daily prednisone, and it could be challenging to withdraw all treatments [12]. According to the retrospective multicenter study by Abisror et al. [12], analyzing 30 patients with EAE, steroid-dependency was frequent, and almost half of the patients required second-line treatment [12].

For these reasons, a specific-steroid sparing treatment is needed for EAE and, over the years, other therapeutic strategies have been performed by many Centers (Table 5).

Interferon-α and cyclosporine-A administration have both been reported, with controversial results [15]. In addition, a therapeutic trial of intravenous immunoglobulin (IVIG) has been tried in a diabetic patient to decrease his risks related to long-term steroid use [17]. Scranton et al. [4] reported an EAE case successfully treated with imatinib mesylate (100 mg/day orally), even if the patient did not show any imatinib-sensitive mutant fusion kinase targets.

In a recent study, an anti-IgE antibody (omalizumab; monthly, 150 mg, subcutaneously) was administered for six months in a patient diagnosed with EAE suffering from frequent AE attack and elevated eosinophils count despite maintenance therapy with glucocorticoids. Since the beginning of treatment with omalizumab, there was no evidence of recurrence during the eighteen-month follow-up period after stopping all medications, including the systemic corticosteroid [31].

Mepolizumab, a fully-humanized monoclonal antibody that targets IL-5, has been approved to treat severe eosinophilic asthma, eosinophilic granulomatosis polyangiitis, and HES [43,44]. Because mepolizumab has been demonstrated to be safe and effective in patients with *FIP1L1-PDGFRA* negative HES at the dose of 750 mg administrated intravenously every four weeks, Matucci et al. [15] used the same mepolizumab dose in a thirty-seven-year-old man diagnosed with EAE, who was successfully treated. The overproduction of IL-5 is an essential determinant of the pathophysiology of EAE because its serum level is elevated during attacks [13]. Regulating this pathway may be useful for modulating inflammatory response induced by sustained activation of eosinophils [45].

A Phase II currently ongoing trial (NCT04128371) is evaluating the impact of mepolizumab administration on frequency and severity of symptoms and on the initial and sustained reduction in the blood eosinophil count in 12 adult subjects with EAE. Subjects received three administrations of mepolizumab 700 mg intravenously, one infusion every 4–6 weeks. The full details of the trials are available at the website www.clincaltrials.gov, accessed on 15 March 2021.

The recent growing number of biological therapies targeting eosinophils allows a good expectation for the development of targeted therapies to treat this rare eosinophilic disorder. For this reason, further randomized controlled clinical trials, together with a better understanding of EAE pathogenesis, are needed for identifying steroids-sparing therapeutic approaches to provide a safe and effective target treatment to manage this rare disease.

## Figures and Tables

**Figure 1 jcm-10-01442-f001:**
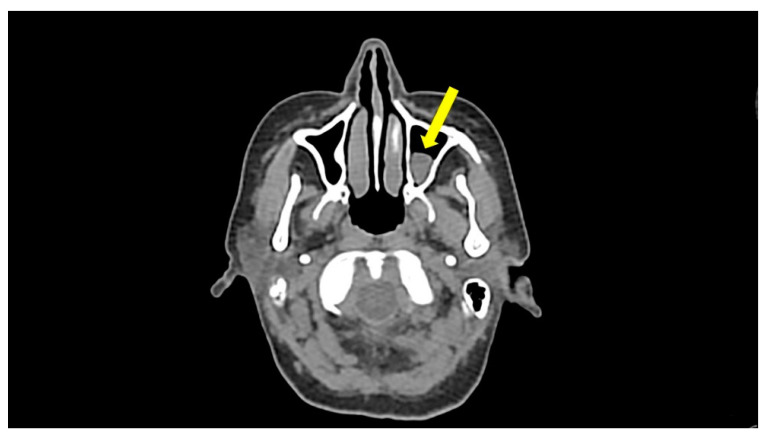
Noncontrast axial maxillofacial computed tomography scan in a patient with episodic angioedema with eosinophilia. The inferior portion of the left maxillary sinus lumen is filled with fluid inflammatory material (yellow **arrow**). No bony erosion of the wall of the sinus is appreciated.

**Figure 2 jcm-10-01442-f002:**
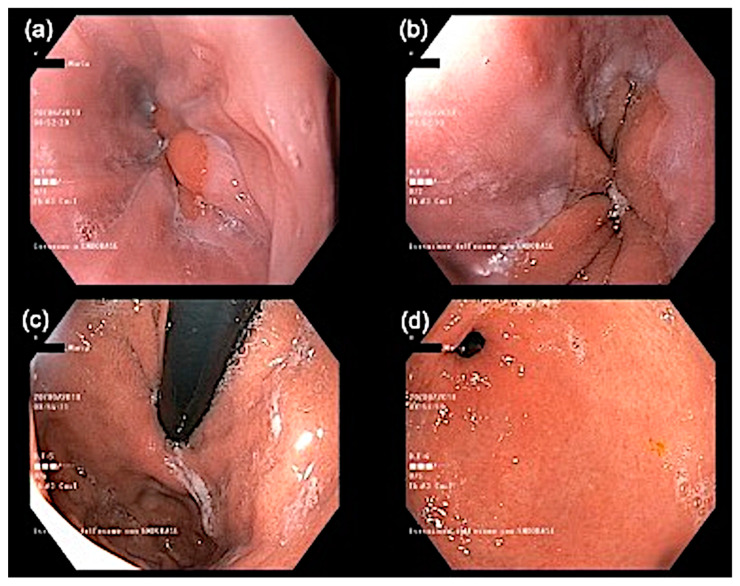
Upper gastrointestinal endoscopy in a patient with episodic angioedema with eosinophilia. The squamocolumnar junction “Z-line” is displaced cranially, which is suggestive of sliding hiatal hernia; incontinence of the cardia is observed (**a**–**c**). The presence of nonconfluent mucosal breaks that are ≤5 mm in length (**d**) revealed mild esophagitis.

**Figure 3 jcm-10-01442-f003:**
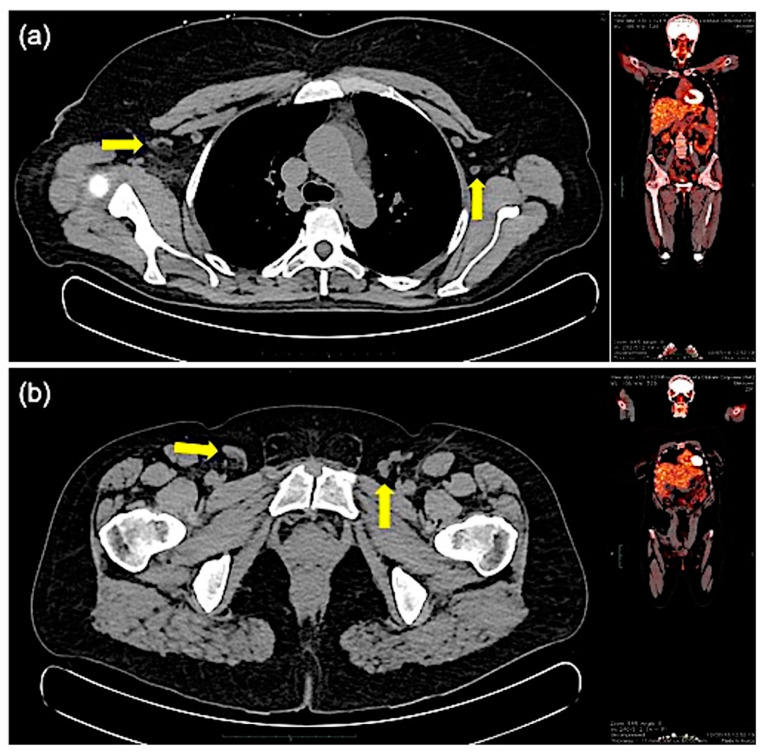
Positron Emission Tomography/Computed Tomography (PET/CT) total body scan. The exam revealed multiple cervical, mediastinal, and axillary (yellow arrows) lymphadenopathy (maximum diameter 1.3 cm) in the absence of fluoro-D-glucose (FDG) uptake (**a**). Reactive lymphadenopathies (yellow arrows) were also detected at the examination of the inguinal regions (**b**).

**Figure 4 jcm-10-01442-f004:**
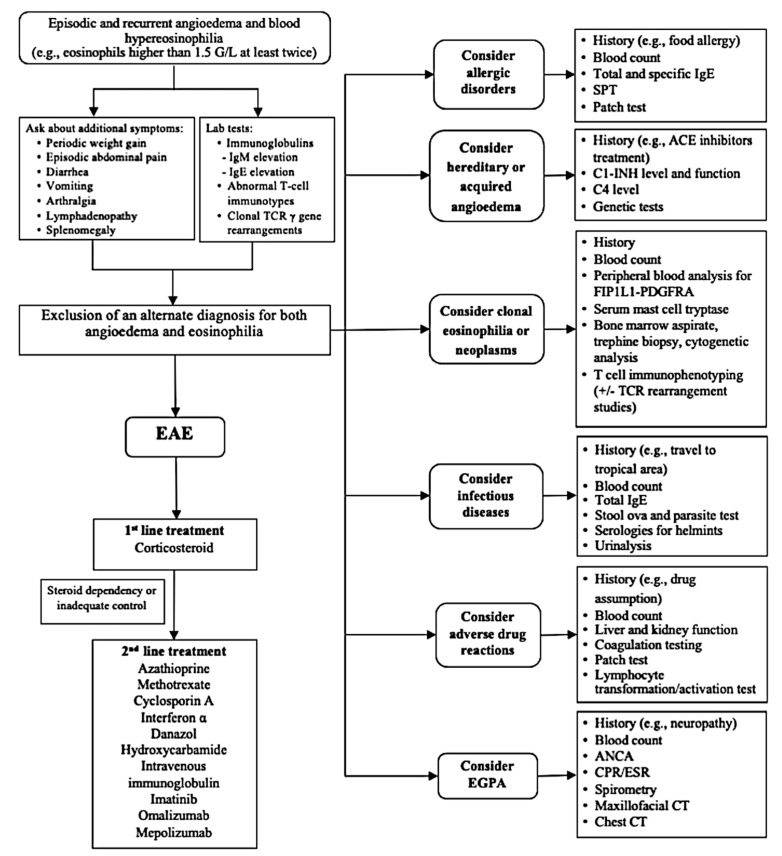
Flow diagram of diagnosis and treatment of episodic angioedema with eosinophilia (EAE). TCR: T-cell receptor; Ig: immunoglobulin; SPT: skin prick test; ACE: angiotensin-converting enzyme; C1-INH: C1-inhibitor; ANCA: anti-neutrophil cytoplasmic antibodies; CPR: C-reactive protein; ESR: erythrocyte sedimentation rate; CT: computed tomography; EGPA: eosinophilic granulomatosis with polyangiitis.

**Table 1 jcm-10-01442-t001:** Characteristics of the studied patient.

	Patient’s Data	Normal Range
Gender	Female	-
Age of onset (years)	30	-
Age at diagnosis (years)	32	-
Symptoms	Angioedema Urticaria Pruritus Periodic weight gain Gastrointestinal symptoms Splenomegaly Lymphadenopathy	-
AEC (g/L)	9.617	0–0.45
ANC (g/L)	9.009	1.8–7
Serum total IgM (g/L)	2.080	0.40–2.30
Serum total IgG (g/L)	9.410	7.37–16.07
Serum total IgA (g/L)	1.320	0.70–4.00
Serum total IgE (KU/L)	30.90	<100
Aberrant T-cell population	No	-
ESR(mm/Ih)/CRP(mg/dL)	10/ < 0.33	0–15/0–0.50
ANA (Index)	0.11	0–1
ANCA-MPO(UI/mL)	0	<10
PR3-ANCA(UI/mL)	0	<10
C3 (g/L)	1.28	0.9–1.8
C4 (g/L)	0.27	0.1–0.4
C1-inhibitor (g/L)	0.292	0.21–0.39
C1q (mg/dL)	19	10–25
Maxillofacial CT	Unilateral maxillary sinusitis	-
Upper GI endoscopy	Mild esophagitis	-
GI histology	No eosinophilic infiltrate	
Treatment	Prednisone	-
Outcome	Remission	-

AEC: absolute eosinophil count at the onset; ANC: absolute neutrophil count at the onset; ANA: anti-nuclear antibodies; ANCA: anti-neutrophil cytoplasmic antibodies; CPR: C-reactive protein; CT: computed tomography; ESR: erythrocyte sedimentation rate; GI: gastrointestinal.

**Table 2 jcm-10-01442-t002:** Episodic angioedema with eosinophilia cases reported in the literature and main clinical features.

Symptoms	Number of Patients (%) Number of Studied Patients = 57	References
Angioedema	57 (100%)	[2,3,4,5,7,12,13,14,15,16,17,18,19,20,21]
Weight gain	27 (47.3%)	[2,3,5,7,12,13,14,15,16,17,18,19,20,21]
Urticaria	23 (40.3%)	[2,3,5,12,19,20]
Gastrointestinal symptoms	15 (26.3%)	[5,7,12,18]
Pruritus	14 (24.5%)	[12,14,17,20]
Arthralgia	10 (17.5%)	[12,17,22,23]
Splenomegaly	9 (15.7%)	[2,12,20]
Fever	9 (15.7%)	[2,4,7,14,15,17,19]
Lymphadenopathy	8 (14%)	[12,16]
Eosinophilic myocarditis	2 (3.5%)	[12]
Asthenia	1 (1.7%)	[20]
Oliguria	1 (1.7%)	[15]
Myalgias	1 (1.7%)	[15]
Ascites	1 (1.7%)	[18]
Arterial hypotension	1 (1.7%)	[20]
Sore throat	1 (1.7%)	[17]
Dyspnea on exertion	1 (1.7%)	[7]

**Table 3 jcm-10-01442-t003:** Most commonly reported laboratory findings in episodic angioedema with eosinophilia.

	Range Value [Number of Patients (%)] Number of Studied Patients = 57	References
AEC(g/L) ^1^	0.88–95 [57 (100%)]	[2,3,4,5,7,12,13,14,15,16,17,18,19,20,22]
ANC (g/L) ^2^	1.4–15.6 [11 (19.2%)]	[1,3,4]
	NR [46 (80.7%)]	
IgM levels	normal [17 (29.8%)]	[2,3,4,5,7,12,13,14,15,16,17,18,19,20,22]
	high [38 (66.6%)]	
	NR [2 (3.5%)]	
IgE levels	normal [28 (49.1%)]	[2,3,4,5,12,13,14,15,16,19,20,22]
	high [26 (45.6%)]	
	NR [3 (17.1%)]	
T cell populations	normal [21 (36.8%)]	
	aberrant [18 (31.5%)]	[3,5,7,12,15,20]
	NR [18 (31.5%)]	

^1^ AEC: absolute eosinophil count (reference range: 0–0.45 g/L); ^2^ ANC: absolute neutrophil count (reference range: 1.8–7 g/L); NR: not reported.

**Table 4 jcm-10-01442-t004:** Episodic angioedema with eosinophilia differential diagnosis.

**A.** **Allergic disorders** Urticaria Atopic dermatitis/eczema
**B.** **Primary (clonal) eosinophilia** Hematological neoplasms accompanied by an eosinophilia in which eosinophils are part of the neoplastic clone
**C.** **Idiopathic eosinophilia** No detectable primary or secondary causes for eosinophilia
**D.** **Neoplasms** Solid tumors Lymphomas and acute lymphoblastic leukemia Systemic mastocytosis Lymphoid-HES
**E.** **Infectious diseases** Parasitic infections Fungal infections
**F.** **Adverse drug reactions** Drug rash with eosinophilia and systemic symptoms (DRESS)
**G.** **Miscellaneous conditions** Hereditary Angioedema Acquired Angioedema IL-2-induced capillary leak syndrome Eosinophilic granulomatosis with polyangiitis (EGPA) (Churg-Strauss syndrome)

**Table 5 jcm-10-01442-t005:** Episodic angioedema with eosinophilia cases reported in the literature and treatment strategies.

Treatment	Number of patients (%) Number of Studied Patients = 57	References
Corticosteroids	52 (91.2%)	[2,3,4,5,12,13,14,15,16,17,18,19,20,22]
Azathioprine	1 (1.7%)	[15]
Methotrexate	4 (7%)	[12]
Cyclosporin A	6 (10.5%)	[12,15]
Interferon α	9 (15.7%)	[5,12]
Danazol	1 (1.7%)	[12]
Hydroxycarbamide	2 (3.5%)	[12]
Intravenous immunoglobulin	1 (1.7%)	[17]
Imatinib	2 (3.5%)	[4,12]
Omalizumab	1 (1.7%)	[13]
Mepolizumab	1 (1.7%)	[15]

## Data Availability

The data presented in this study are available in within the article.

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
