# Peer review of "Episodic Angioedema with Hypereosinophilia (Gleich’s Syndrome): A Case Report and Extensive Review of the Literature"

_jcm, 2021, doi:10.3390/jcm10071442_

Round 1
Reviewer 1 Report
The authors reported a clinical case and literature review on Gleich’s syndrome. There are already many reviews on this topic. The paper adds nothing new. This paper is similar: Haber R, Chebl JA, El Gemayel M, Salloum A. Gleich syndrome: a systematic review. Int J Dermatol. 2020 Dec;59(12):1458-1465. doi: 10.1111/ijd.14963. Epub 2020 Jun 18. PMID: 32557651.
In my opinion, the authors, starting from the clinical case, could propose a flowchart with a diagnostic and therapeutic approach to disease.
Can't add a photo in the case description "unilateral periorbital edema of the facenwas evident on physical examination"?
When they talk about therapies, they should mention the ongoing trials (NCT04128371)
Reviewer 2 Report
Dear Authors,
I read with great interest Your manuscript.
Gleich's syndrome is a very rare syndrome, and Your case report should be taken notice of.
However, my personal opinion is that it needs extensive revision, before it can be accepted for publication.
Here are my comments and suggestions. I hope they are useful.
1) first of all, Yours was a case-based review (and not a review);
2) the structure of article should be reviewed. Indeed, You should first report the case and then evaluate a literature search (please, see paragraph 2 and modify it in line with this suggestion);
3) please, consider a Table where all medical data of patient are listed;
4) specify how long was her follow-up;
5) in references, I did not found Haber R et al, doi: 10.1111/ijd.14963. Please, consider to add it;
6) lines from 185 to 198 were unclear. I suggest to rewritten them;
7) as You highlighted, Gleich's syndrome "resolve spontaneously without therapy" (see, lines 20-21, line 38 and 79). On the other hand, Your patients was treated with high doses of oral glucocorticoids. Can You clarify this apparent contradiction ? Thanks.
Minor comment : C-reactive protein (CRP) concentrations (see line 104).
Round 2
Reviewer 1 Report
I am satisfied with the corrections made
Reviewer 2 Report
Dear Authors,
I read the newer, revised version of Your manuscript.
All my comments and suggestions were satisfactorily met. No doubt.
I hope that the scientific soundness and the quality of presentation have consequently improved.